# 3D ocean assessments reveal that fisheries reach deep but marine protection remains shallow

Juliette Jacquemont [1,2] ✉, Charles Loiseau[2], Luke Tornabene[1] &
Joachim Claudet [2] ✉

The wave of new global conservation targets, the conclusion of the High Seas Treaty negotiations, and the expansion of extractive use into the deep sea call for a paradigm shift in ocean conservation. The current reductionist 2D representation of the ocean to set targets and measure impacts will fail at achieving effective biodiversity conservation. Here, we develop a framework that overlays depth realms onto marine ecoregions to conduct the first three-dimensional spatial analysis of global marine conservation achievements and fisheries footprint. Our novel approach reveals conservation gaps of meso-photic, rariphotic, and abyssal depths and an underrepresentation of high protection levels across all depths. In contrast, the 3D footprint of fisheries covers all depths, with benthic fishing occurring down to the lower bathyal and mesopelagic fishing peaking in areas overlying abyssal depths. Additionally, conservation efforts are biased towards areas where the lowest fishing pressures occur, compromising the effectiveness of the marine conservation network. These spatial mismatches emphasize the need to shift towards 3D thinking to achieve ocean sustainability.

Global conservation efforts are about to significantly expand as the United Nations (UN) Convention on Biological Diversity (CBD)'s Kunming-Montreal Global Biodiversity Framework (GBF) has set the course to cover 30% of land and ocean with area-based conservation tools by 2030[1]. In addition, the GBF recognizes that both protected areas and other effective area-based conservation measures (OECMs) can contribute towards area-based conservation targets. OECMs are geographically defined areas that, unlike protected areas, do not have biodiversity conservation as a primary objective, but still achieve bio-diversity benefits from their management[2]. In addition, a legally binding instrument under the UN Convention on the Law of the Sea to protect and sustainably use marine biological diversity in areas beyond national jurisdiction (BBNJ), otherwise known as the High Seas Treaty, has just been concluded. These agreements bring considerable opportunities for marine conservation by increasing coverage targets,

by diversifying the types of governance regimes and sectors that can contribute to area-based conservation, and by vastly extending areas that can be conserved. However, they also create new challenges for conservation planning, such as designing area-based conservation tools suited for off-shore, deep, and vertically complex areas[3,4], as well as mindfully incorporating OECMs that are often vertically zoned in the conservation network[2]. Realizing the potential of these agreements requires a shift from a reductionist two-dimensional (2D) representation of the ocean to a three-dimensional (3D) representation of eco-systems, human use, and impacts[5–7]. This shift is essential to avoid compounding on the weaknesses of the current conservation network, which already fails at achieving ecological representation[8], high levels of protection[9], and at abating human impacts[10].

The ocean is inherently three-dimensional. Unlike on land, life in the ocean spans over a considerable vertical range from the surface to

[1]School of Aquatic and Fishery Sciences, University of Washington, 1122 NE Boat St, Seattle, WA, USA. [2]National Center for Scientific Research, PSL Université Paris, CRIOBE, CNRS-EPHE-UPVD, Maison de l'Océan, 195 rue Saint-Jacques, Paris, France. ✉e-mail: juliette.jacquemont.fr@gmail.com; joachim.claudet@cnrs.fr

the seafloor, with an average depth of 3800 m. However, apart from recent model-based prioritization studies[11–13], assessments of human use[14,15] and of conservation achievements[8,16] remain two-dimensional. Although the vertical stratification of marine life and human use has long been recognized, multiple factors have led to the persistence of 2D representations. The historical predominance of terrestrial conservation and of land-use management have shaped marine conservation and marine spatial planning, with human use and UN CBD conservation targets mostly allocated in 2D. This approach has remained mostly unchallenged because human activities, scientific research, and conservation have historically been constrained to shallow marine environments where vertical structure is simple[17]. Besides, the fragmented nature of ocean governance, with multiple sector- and area-specific regimes, hampers a holistic three-dimensional management of the ocean[18].

Distinct scientific groups have previously raised awareness on the fact that deep marine ecosystems, such as the mesopelagic, deep reefs, and seamounts, are under increasing human pressures and require dedicated conservation efforts[19–21]. However, a three-dimensional framework to identify conservation gaps and priority areas *across* depths is still missing. In particular, while fisheries are the main driver of marine biodiversity erosion[22], global assessments of fisheries' footprint remain two-dimensional[14,23–25] and fail to inform on which depths are being targeted[26–28]. This represents a critical knowledge gap to inform fisheries management and marine conservation because the sensitivity of ecosystems to fishing pressure varies greatly with depth[29]. Global 2D mapping of marine cumulative impacts[14,30] have been instrumental in guiding marine policies by demonstrating the extent and acceleration of the human footprint on the ocean. There is now a need for 3D

mapping of the human footprint across space and depth to guide global conservation policy.

Here, we develop a novel mapping framework by overlaying benthic and pelagic depth realms, which captures main ecological units across depths, onto commonly used 2D marine ecoregions[31,32]. Using this mapping framework, we assess for the first time the 3D ecological representativeness of the global ocean conservation network and conduct the first global assessment of the depth distribution of fishing effort. We then test whether marine protected areas (MPAs) and OECMs are appropriately sited to provide protection to areas under high fishing pressure across both space and depth. Based on our results, we identify conservation gaps, conservation priorities, and provide recommendations on how to account for specificities linked to the three-dimensionality of the ocean in the global conservation agenda.

## Results and discussion
### Uneven conservation effort across depth
Based on the vertical zonation of marine species, habitats, and environmental conditions[33–35], we identified eight benthic and four pelagic realms that represent ecological units across depths (Fig. 1).

We found that the distribution of protection coverage (MPAs and OECMs) is uneven across depth realms (Figs. 2 and S3) and ecoregions (Fig. 3). The euphotic (0 to −30 m) is the best protected depth realm, combining the second largest protection coverage (15%, Fig. 2B), the second largest coverage of Ia and Ib protected area categories of the International Union for Conservation of Nature (IUCN) (1.2%), which tend to more strictly regulate human use (see Table S1 for details on IUCN categories), and the most widespread protection across ecoregions (Fig. 3 and S5). By contrast, the abyssal realm has the smallest

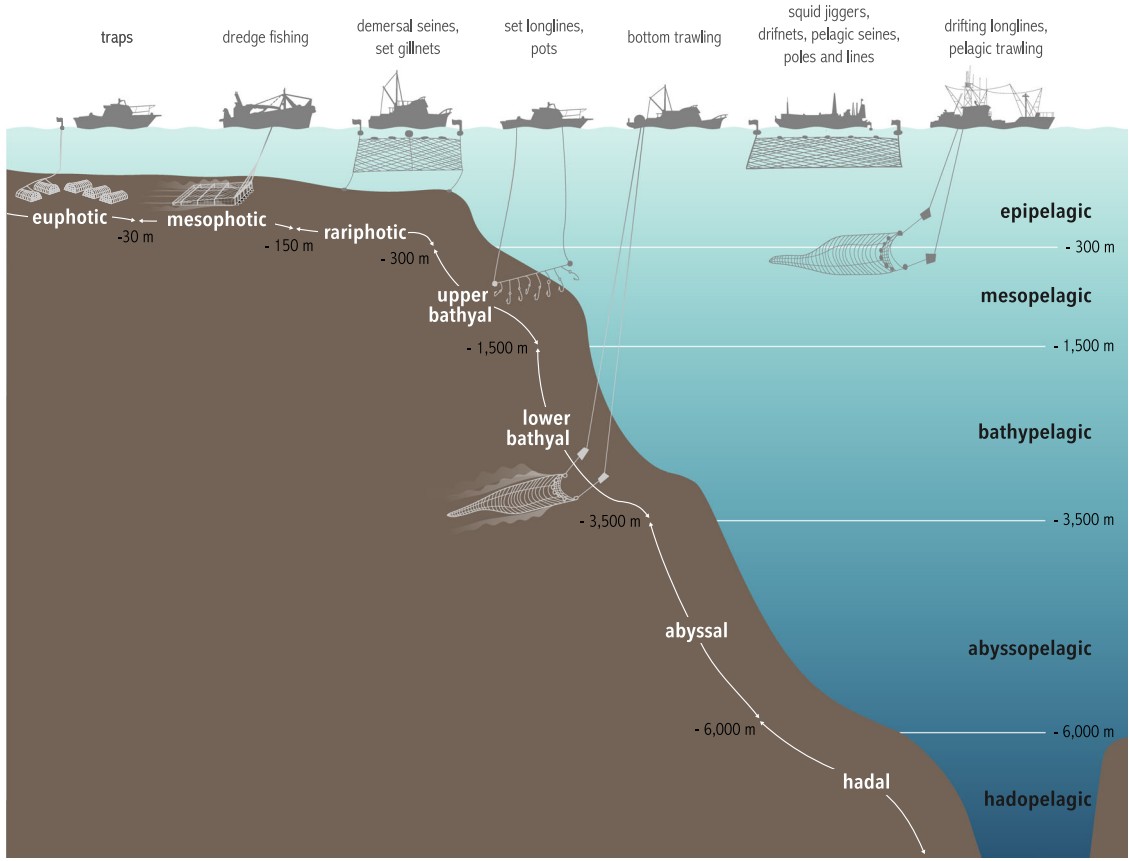

**Fig. 1 | Vertical distribution of benthic realms, pelagic realms, and depths targeted by fishing gear.** The depth at which fishing gears are represented indicates the maximal depth at which these gears are operated. Only one type of fishing gear was depicted per depth maxima, but the same depth maxima apply to all gears listed in a same column (e.g., set longlines and pots are deployed down to the upper bathyal). The mesophotic benthic realm (−30 to −150 m) was further subdivided into "upper mesophotic" (−30 to −60 m) and "lower mesophotic" (−60 to −150 m) in our analyses.

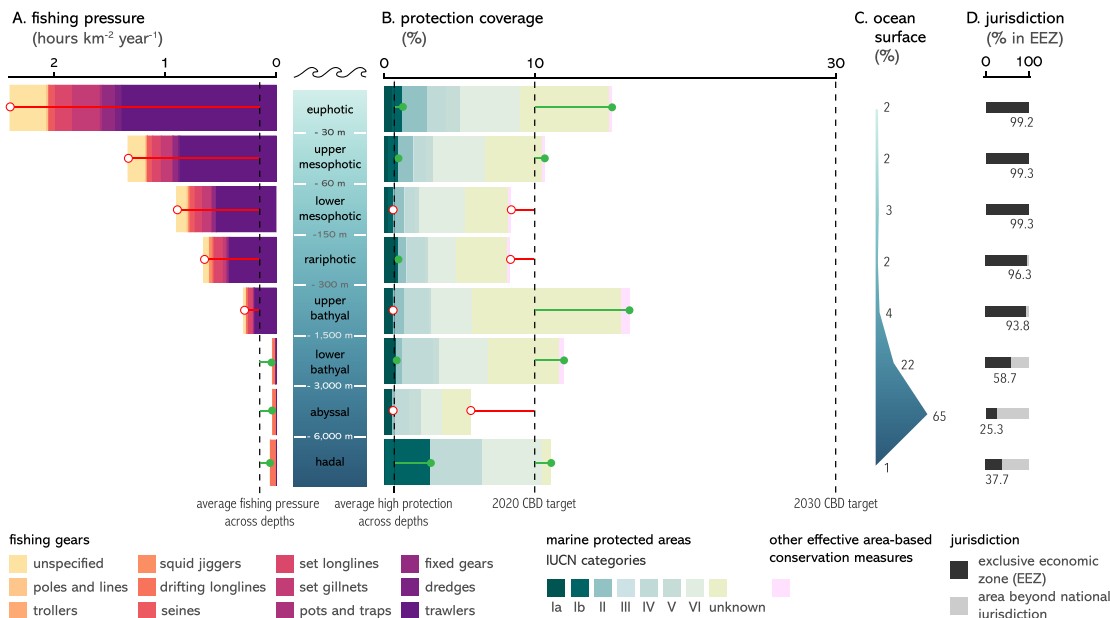

**Fig. 2 | Distribution of fishing pressure and conservation efforts across depth realms. A** Average fishing pressure by fishing gear across depth realms. Lollipops indicate whether fishing pressure in each depth realm is above (red lollipops) or below (green lollipops) global average fishing pressure. **B** Protection coverage of marine protected areas (MPAs) by IUCN categories and other effective area-based conservation measures (OECMs) across depth realms. Lollipops indicate whether the current protection coverage of depth realms is behind (red lollipops) or ahead (green lollipops) of the average coverage of high protection and of the 2020 CBD target. **C** Proportion of the ocean falling under each depth realm. **D** Proportion of depth realms falling under exclusive economic zones or areas beyond national jurisdiction. The four vertical dashed lines represent from left to right: average fishing pressure across depths, average coverage of high protection (MPAs of Ia and Ib IUCN categories) across depths, and the 2020 and 2030 CBD coverage targets.

extent of protection coverage (5.8%) and the smallest coverage of Ia and Ib IUCN categories (0.6%). The BBNJ treaty now provides the legislative framework to increase abyssal protection coverage as 75% of this depth realm occurs in areas beyond national jurisdiction (Fig. 2D). When considering depth realms predominantly occurring within economic exclusive zones (EEZs), lower mesophotic and rariphotic realms are the least protected globally (Fig. 2B) as well as across most of the world's coastal ecoregions (extending 200 nautical miles offshore, Fig. 3).

Across all depths, the majority of protection coverage falls under IUCN category VI or unknown (Fig. 2B), which correspond to the lowest levels of protection from human use[36]. Most 3D realms are covered by less than 1% of Ia and Ib IUCN categories (0.7% as a global average) and one-third by less than 0.1% (Fig. 3), which correspond to the highest levels of protection. The greatest coverage of Ia and Ib IUCN categories is found in the hadal realm (3%), in the Eastern-Indo Pacific (10%), and in polar ecoregions (Arctic and Southern Ocean, 1–2%).

### Distribution of fishing activities across space and bathymetry

We assessed the 3D distribution of fishing footprint by overlaying the spatial distribution of fishing activities reported by the Global Fishing Watch (GFW[33]) with the highest resolution bathymetric map of the ocean[37]. We found that both fishing pressure and fishing gears are highly structured by bathymetry (Fig. 2A). Areas overlying euphotic to upper bathyal depths experience above average fishing pressure dominated by trawlers, while areas overlying lower bathyal to hadal depths experience lower fishing pressure dominated by drifting longline fisheries (Fig. 2A). Average fishing pressure (hour km$^{-2}$ year$^{-1}$, Fig. 2A) and total fishing effort (hours year$^{-1}$, Fig. 4) are highest in the euphotic. Fishing pressure continuously decreases in areas overlying greater depths, with a sharp six-fold decrease in areas deeper than 1500 m (upper bathyal). However, total fishing effort is as high in areas overlying abyssal depths as in areas overlying mesophotic depths. In ecoregions where average fishing pressure is the highest

(Temperate Northern Atlantic and Pacific, Temperate South America), average fishing pressure remains high from areas overlying euphotic to upper bathyal depths, while in off-shore ecoregions and in some coastal ecoregions (e.g., Southern Africa, Australasia) average fishing pressure peaks in areas overlying rariphotic to bathyal depths (Fig. 4).

Although bathymetry alone is insufficient to determine the depths targeted by fishing activities, it can inform on the broader depth range likely impacted by fishing pressure. These indirect impacts occur through vertical connectivity processes, such as migration of organisms, top-down trophic controls, or nutrient transfers[38–40]; and through by-catch, entanglements, anchoring, fishing debris, or ship collisions[41]. Concerns are now being raised that mesopelagic fisheries could even affect carbon sequestration in deep-sea sediments by altering the ocean's biological carbon pump[42]. As such, in data-poor contexts, relying on the bathymetric distribution of fishing activities can be a first entry point to assess the 3D distribution of fishing impacts.

### Elucidating the three-dimensional distribution of fishing activities

The disaggregation level of fishing gears reported in the GFW datasets allows to distinguish between pelagic and benthic fisheries for 55% of the total fishing hours reported from 2018 to 2020. Depths targeted by benthic activities can be determined with accuracy as they match the bathymetry of the fishing location. For pelagic fishing activities, we determined depths targeted based on the typical depth range of the fishing gear reported by the GFW and the bathymetry at the fishing location (Fig. 1, Table S3). We found that benthic fishing effort was greatest in the euphotic and mesophotic (>10$^7$ hours year$^{-1}$) but remained important down to the upper bathyal (2 × 10$^6$ hours year$^{-1}$) and occurred down to the lower bathyal (Fig. 4). Most pelagic activities occurred over lower bathyal to abyssal depths and corresponded to mid-water trawls and drifting longlines.

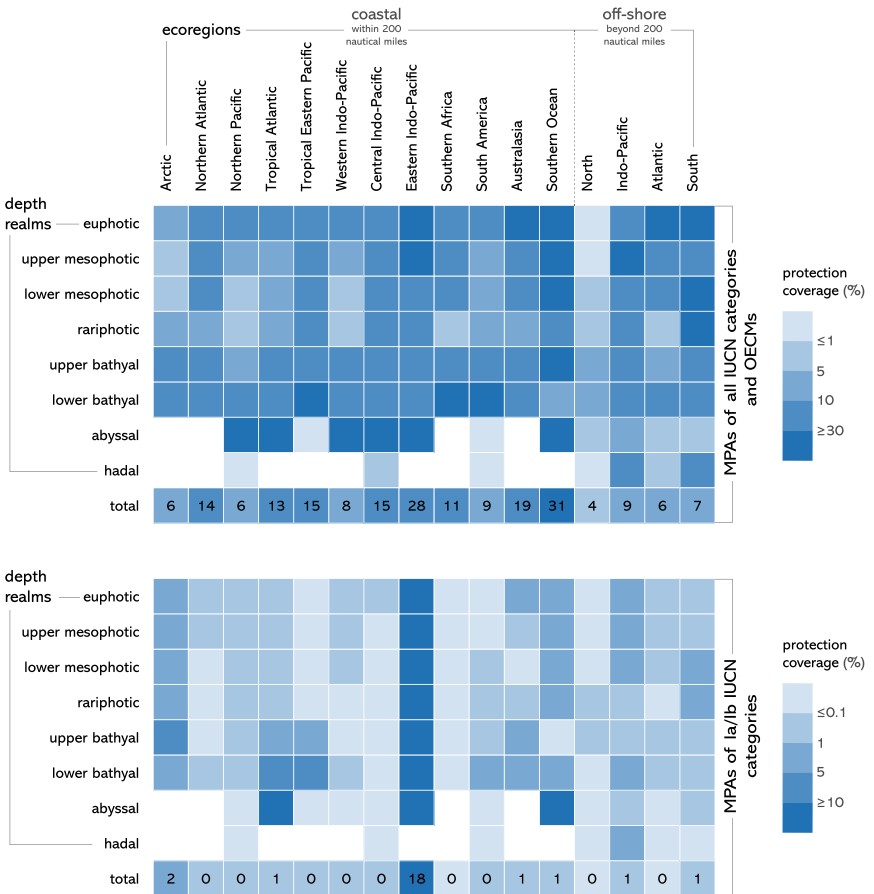

**Fig. 3 | Three-dimensional distribution of marine conservation efforts.** Protection coverage across depth realms per ecoregions for marine protected areas (MPAs) of all IUCN categories and other effective area-based conservation measures (OECMs) (upper panel) and for MPAs of Ia/Ib IUCN categories only (lower panel). Void cells indicate depth realms that do not occur in a given ecoregion. The last row of each panel ('total') represents the total protection coverage (%) across depths in a given ecoregion.

We found that the 3D footprint of fisheries extends across most depths in most ecoregions of the world. In particular, we found that 37% of total fishing effort overlies depths greater than 300 m and thus directly or indirectly impacts deep marine ecosystems (Fig. 4). This is the result of decades of fishing down the deep as shallow and coastal fish stocks have been depleted from overfishing[26–28,43,44]. Unfortunately, in addition to being unprofitable if not heavily subsidized[45,46], deep fisheries are often unsustainable[29,47,48], with high rates of by-catch and long-lasting impact on habitats[49,50]. Some regions have taken action to limit the depth of fishing activities (e.g., trawling ban below 800 m in European waters), but more depth regulations are needed to ensure sustainable fishing practices[27].

Our study is the first attempt to characterize the 3D distribution of fishing activities at the global scale and reflects the limited available knowledge to characterize the 3D fishing footprint. Importantly, about 45% of fishing activities reported in the GFW database do not discriminate pelagic and benthic activities (Fig. 4). Alternatively, datasets that distinguish pelagic and benthic fishing activities, such as those produced by some Regional Fisheries Management Organizations, provide catch data pooled by large spatial units, which also prevents from determining the depth distribution of fishing activities. Systematically distinguishing between pelagic and benthic fishing and increasing the precision of the spatial information associated with catch data would constitute important steps forward to improve our understanding of the depths targeted by fisheries. Our results underestimate fishing pressure overlying euphotic to rariphotic depths because the GFW database only documents vessels with automatic

identification systems, which does not capture most small-scale fisheries, especially in the Caribbean, South-West Pacific, and Indian Ocean where catches are systematically underreported[23,25].

## Mismatched distribution of fishing and conservation efforts

We evaluated whether protection coverage across 3D realms is appropriately sited to mitigate fishing pressure by testing whether MPAs and OECMs are implemented in highly fished realms. We found that the protection coverage of 3D realms (log transformed) was negatively correlated with fishing pressure (Fig. 5), indicating a large bias of ocean conservation towards least impacted areas. This negative correlation was significant when considering coverage by all MPAs and OECMs ($p = 0.021$, $R^2 = -0.22$), and when considering coverage by only MPAs of Ia/Ib IUCN categories ($p < 0.001$, $R^2 = -0.33$).

We defined four profiles of conservation priority (Fig. 6b) based on the fishing pressure (below or above median) and protection coverage of 3D realms (behind or past halfway progress from 2020 to 2030 targets). We found that highest conservation priority areas, i.e., areas with low protection coverage and high fishing pressure, mostly occur in the mesophotic, rariphotic, and upper bathyal realms across all ecoregions of the world (Fig. 6a, b). Lowest conservation priority areas, i.e., areas that combine high conservation coverage and low fishing pressure, mostly occur at lower bathyal and abyssal depths in coastal ecoregions. Implementing future MPAs or OECMs in such low use and widely represented areas would decrease the net ecological benefits achievable by these conservation tools. If areas of low fishing pressure still need to be targeted to minimize impacts on fisheries,

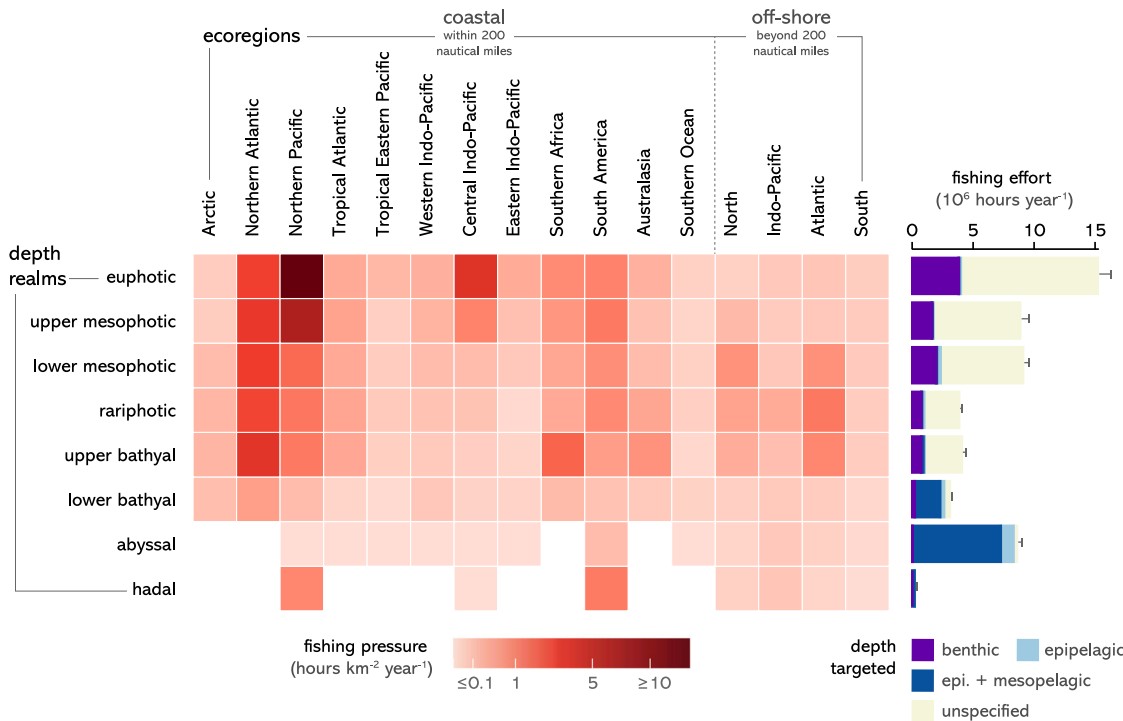

**Fig. 4 | Three-dimensional distribution of fishing pressure.** Average fishing pressure across depth realms and ecoregions (left panel) and absolute fishing effort per benthic and pelagic depth realms (right panel). Void cells indicate depth realms that do not occur in a given ecoregion. Error bars represent 95% confidence interval from n = 3 years of fishing data (2018–2020). The 'unspecified' category indicates ambiguous gear types in the Global Fishing Watch database (e.g., "trawling" without distinction between mid-water and bottom trawling). Note that for fishing pressures over lower bathyal and abyssal waters indicated in the left panel, most fishing occurred in shallower pelagic realms (epipelagic and mesopelagic, 0–1000 m).

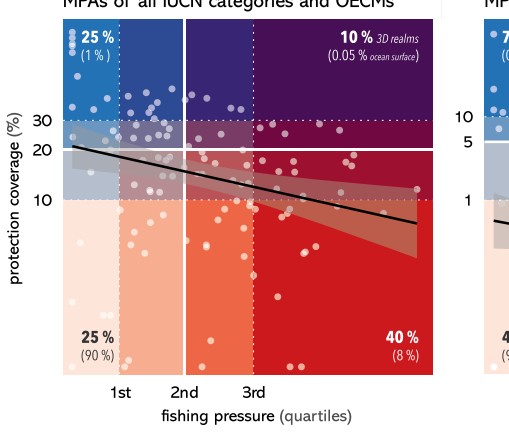

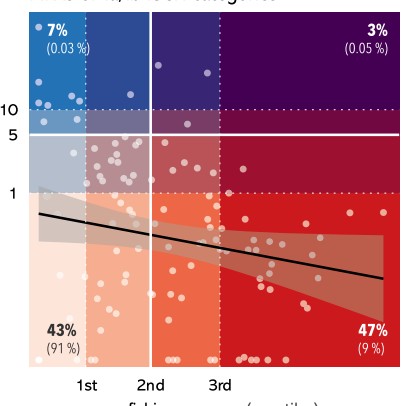

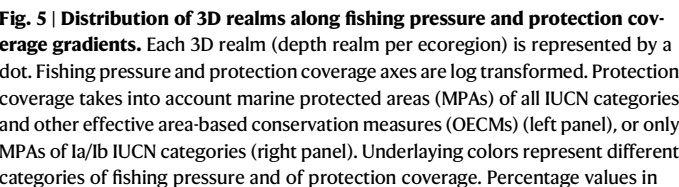

**Fig. 5 | Distribution of 3D realms along fishing pressure and protection coverage gradients.** Each 3D realm (depth realm per ecoregion) is represented by a dot. Fishing pressure and protection coverage axes are log transformed. Protection coverage takes into account marine protected areas (MPAs) of all IUCN categories and other effective area-based conservation measures (OECMs) (left panel), or only MPAs of Ia/Ib IUCN categories (right panel). Underlaying colors represent different categories of fishing pressure and of protection coverage. Percentage values in bold represent the proportion of 3D realms falling into each of the four main categories (below/above median fishing pressure and behind/past halfway progress from 2020 to 2030 conservation targets). Percentage values in parentheses represent the proportion of ocean surface falling under these same categories. Black lines indicate the linear regression model between log-transformed fishing pressure and log-transformed protection coverage, and shaded areas represent the 95% confidence interval associated with that model.

then targeting 3D realms that suffer from conservation representation gaps should be the priority. Such 3D realms mostly occur in lower bathyal and abyssal depths in off-shore ecoregions and will require conservation action in the high seas (Fig. 6b). Implementing and actively managing MPAs of high protection levels in any 3D realm would provide important conservation benefits, as 90% of 3D realms have still not reached 5% of high protection coverage and 47% cumulate low coverage with high fishing pressure (Fig. 5), therefore falling under the highest priority category (Fig. 6b).

Here, we restricted our 3D conservation prioritization to two variables: protection coverage and fishing pressure. While fishing pressure is considered as the main direct anthropogenic threat to marine life[22], other human pressures could be considered to translate our framework into actionable recommendations towards other sectors, especially given the projected rapid expansion of offshore renewable energies, hydrocarbon drilling, and deep-sea mining[19,51,52]. Furthermore, upcoming climate-induced shifts in species[53] and in fishing effort distribution[54,55] should be accounted for to prioritize

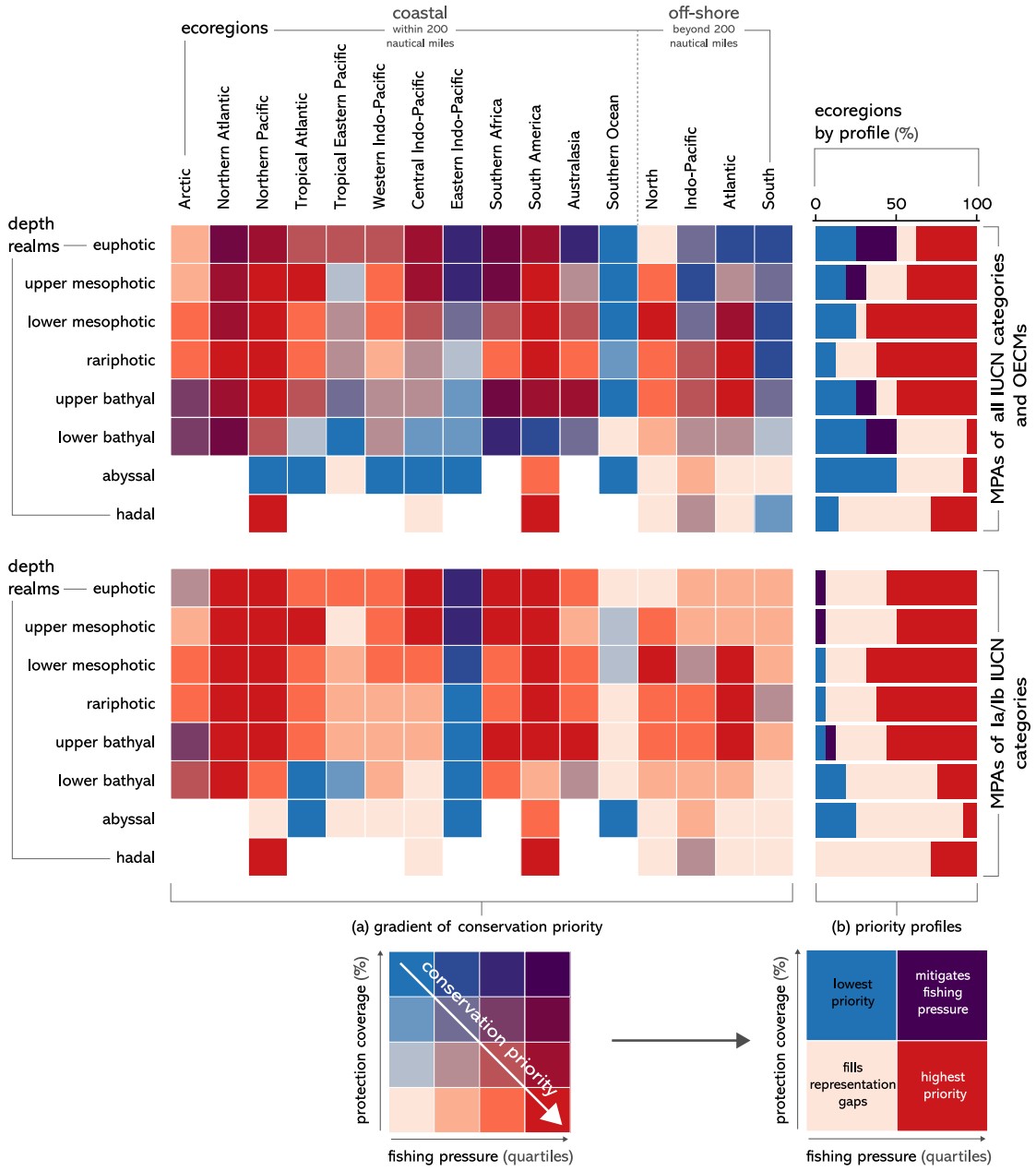

**Fig. 6 | Three-dimensional distribution of conservation priority areas based on fishing pressure and protection coverage.** Distribution of conservation priority profiles across depths based on fishing pressure and progress towards 2030 conservation targets for marine protected areas (MPAs) of all IUCN categories and other effective area-based conservation measures (OECMs) (top left) or only MPAs of Ia and Ib IUCN categories (bottom left). Conservation priority increases with increasing fishing pressure and decreasing protection coverage (**a**). Proportion of conservation priority profiles across depths for all IUCN categories (top right) or only Ia and Ib IUCN categories (bottom right). Four conservation priority profiles were defined based on fishing pressure being below or above median and protection coverage being behind or past halfway completion to 2030 conservation targets (**b**).

climate-smart conservation areas. While species and fishing effort redistribution are unlikely to alter the findings of this study given the large spatial extent of 3D realms, this consideration should be reckoned by studies applying this framework at a finer spatial resolution.

**Towards an even conservation of depth realms**
Several mechanisms can explain the uneven protection of depth realms. The lack of policy tools to designate conservation areas in the high seas has resulted in the near-absence of protection of the abyssal realm, constituting the largest conservation gap of the planet. This gap will start to be addressed when the concluded High Seas treaty will be ratified and the BBNJ COP implemented. Within EEZs, the proximity of

near-shore ecosystems, the incentive to protect areas that can generate tourism revenues[56], and the avoidance of fishing grounds have skewed conservation efforts towards the euphotic or the upper bathyal at the expense of the mesophotic and rariphotic realms. These biases have led to the underprotection of unique but poorly described ecosystems[21,57,58] that are under increasing pressure from human use. Although out of sight, mesophotic and deep ecosystems provide spawning and feeding grounds to valuable fish stocks[59], host the largest amount of undescribed species[17], act as potential climate refugia for shallow species[60], and are central in the ocean carbon cycle[61]. Lastly, fast rates of species and habitat discovery in the mesophotic and in the deep ocean[3] are challenging the perception that only the

euphotic hosts diverse and complex communities. Collectively, these elements demand we revisit current conservation paradigms and extend conservation efforts across all depths to protect the whole range of marine biodiversity.

To address these representation gaps, we propose two paradigm shifts in how conservation efforts are prioritized and measured. First, prioritization studies and measures of conservation representativeness should include criteria suited to data-poor habitats. Current methods to measure biodiversity representation rely on biodiversity features (e.g., species richness, vulnerable habitats, endangered species) that are inherently biased by the more comprehensive description of shallow ecosystems, and thus will keep on disproportionately prioritizing shallow ecosystems (e.g., ref. 62). Because of the scarcity of data on deep habitats and deep species distribution, relying on well-described geophysical features such as depth, which is underway to be fully mapped by 2030[63], can be a relevant strategy to maximize biodiversity representation[64,65]. Such an approach has been demonstrated to be very effective at capturing undescribed biodiversity through incidental representation[66]. In addition, depth representation can represent a climate-smart tool for adaptive conservation planning by acting as a portfolio strategy, and by guaranteeing that species undergoing depth shifts remain in well-conserved habitats[67]. Second, indicators used to track the ecological representativeness of marine conservation networks should include a depth dimension in addition to commonly used 2D units, such as ecoregions[16]. The 3D ecoregion typology (depth realm × ecoregion) developed in this study could serve as the framework to set goals and track progress towards 3D ecological representativeness. The number of depth realms considered and their depth limits could be further adjusted at the regional scale to account for local ecological specificities.

In addition to the uneven distribution of protection coverage, our analysis highlights other flaws of the conservation network that compromise its effectiveness. The correlation between low fishing pressure and high protection coverage (Fig. 5) would ideally be the result of strong regulations that limit fishing pressure. However, it is rather the symptom of residual conservation by which protected areas are placed where they least interfere with human use[10,68]. Furthermore, we found that IUCN categories that dominate the conservation seascape (categories ≥ IV) are those that tend to correspond to lower protection levels[36], although these two classifications are not fully equivalent[69]. Because high or full levels of protection (akin to Ia and Ib IUCN categories), under which extraction is forbidden or strictly regulated, provide the largest ecological, social, and climate benefits[70–72], a target of 10% of high or full protection coverage has been recommended by the scientific community and is already part of the 2030 European Union Biodiversity Strategy[73]. Here, we show how far we remain from this target, as high or full protection covers less than 0.7% of the ocean, is not evenly distributed across depths realms and ecoregions, and is mostly implemented in areas where little fishing pressure occurs.

### Risks of vertically zoned conservation

The proposition of stratifying conservation effort by depth[11,74] has gained momentum with the increasing recognition of area-based fisheries management as OECMs, which commonly only confer protection to the benthos[2]. This assumes that human impacts remain compartmentalized to the depths where human uses occur, which overlooks numerous connectivity processes across depths. The complex energy, nutrient, and population exchanges from the epipelagic to the benthos[75,76] imply that disturbing one depth realm will likely have cascading effects on other depths of that system[40]. This is true for shallow but also for most deep ecosystems, which almost exclusively rely on the biomass and productivity originating from epi- and mesopelagic realms[6,38]. The vertical zonation of conservation tools will thus inevitably result in low levels of protection because some parts of the water column remain exploited, in turn compromising the ability to

yield strong conservation benefits at any depth. Furthermore, the difficulties associated with the enforcement of varying regulations across depths would likely undermine the efficiency of vertically zoned MPAs or OECMs[75]. Finally, vertically-zoned conservation might give yet another gateway for residual conservation. Indeed, it could allow to only protect depths that are not exploited for human use, whether it be the water column above mining sites, or the benthos beneath offshore windfarms. This will make the progress towards biodiversity conservation net gains even harder to measure. To avoid this loophole, we recommend (1) surface-to-seafloor protection as the standard for area-based conservation and (2) if implemented, vertically zoned MPAs and OECMs to be reported with the level of protection of the least protected depth zone.

A two-dimensional representation of the ocean could inform conservation in a world where human use was restricted to shallow ecosystems. However, with the rapid expansion of human use deeper in the ocean and further offshore[52] and the rise of vertically-zoned conservation tools, there is more than ever an urgent need to account for the ocean's three dimensions for conservation planning and regulation of human use. In the wake of a global push for more and better ocean conservation to reach Kunming-Montreal GBF targets, the full three-dimensional range of marine biodiversity needs to be represented in proposed conservation networks. Such considerations are extremely important for national conservation strategies, as 75% of the world's EEZs consist of deep ecosystems while concentrating the highest levels of fishing pressure. In parallel, the recent conclusion of the High Seas treaty offers a unique opportunity to build upcoming conservation efforts on a revised framework that accounts for the complex and intricately connected three-dimensional ocean space.

## Methods

### Definition of three-dimensional marine realms

We obtained a 3D zonation of the ocean by overlaying a 2D zonation of marine ecoregions (e.g., Temperate Northern Atlantic, Tropical Eastern Pacific) with a depth zonation of the main benthic and pelagic realms (e.g., epipelagic, mesopelagic). To define a 2D zonation of marine ecoregions (latitude and longitude), we used the global map of pelagic and coastal realms as described in Spalding et al. (2012, 2007)[31,32]. Realms (hereafter ecoregions) provide the largest spatial unit with coherent biota at high taxonomic levels resulting from shared environmental conditions and evolutionary history[32]. This makes ecoregions relevant units to assess biodiversity representation. Ecoregions are divided into 11 coastal ecoregions and four "pelagic" ecoregions (Fig. S1). Coastal ecoregions extend to 200 nautical miles (370 km) offshore or to the 200-m isobath where the later occurs further. As such, coastal ecoregions cover all waters shallower than 200 m, but also areas of bathyal and abyssal depths when the later occur within 200 nautical miles from shore. "Pelagic" ecoregions cover off-shelf pelagic waters, including waters beyond national jurisdiction. Because all ecoregions actually include both pelagic and benthic ecosystems, we referred to "pelagic" ecoregions as "off-shore" ecoregions hereafter to avoid confusion with the distinction between the benthos versus the water column. We used the ecoregions vector layer from the UN-WCMC Ocean Data Viewer[77] with a precision of 0.01 degrees.

We defined the zonation of benthic depth realms based on the description of ecological depth zones in the literature[33–35,78] as follows: euphotic (0–30 m), upper mesophotic (30–60 m), lower mesophotic (60–150 m), rariphotic (150–300 m), upper bathyal (300–1000 m), lower bathyal (1000–3500 m), abyssal (3500–6000 m), and hadal (below 6000 m). Similarly, we defined the pelagic depth realms as follows: epipelagic (0–200 m), mesopelagic (200–1000 m), bathypelagic (1000–3500 m), abyssopelagic (3500–6000 m), and hadopelagic (below 6000 m).

## Three-dimensional distribution of MPAs and OECMs

We used the World Database on Protected Areas (WDPA[79]), the most comprehensive source on designated protected areas, to build our global map of MPAs[80]. Similarly, we used the World Database on Other Effective Conservation Measures[79] to build our global map of OECMs. We followed the methodology recommended in Thomas et al.[80] to process the WDPA vector layer in a way that generates reliable information on the area and protection level of MPAs. Only MPAs for which spatial boundaries were known were included in the analysis. Although circular buffers have been used in the past to include MPAs for which size but not shape is known (e.g., Spalding et al.[81], we chose not to do so to avoid false information on the bathymetric coverage of MPAs[82]. Terrestrial parts of MPAs were excluded by clipping MPA boundaries with a terrestrial land vector (Natural Earth version 5.1.1, 10-m resolution). The MPA vector layer was simplified to a resolution of 0.01 degree to save memory consumption and computation time.

To account for the MPAs' level of protection[70] we used the management categories defined by the International Union for Conservation of Nature (IUCN)[83] from the WDPA (Table S1). Although the IUCN categories and the levels of protection as defined by the MPA guide do not have perfect correspondence[36], the IUCN management categories reflect a gradient from exclusive biodiversity protection (Ia) to integrated human use and extraction (VI), which we used here as a proxy for levels of protection. IUCN categories reported as "Not Applicable", "Not Reported" or "Not Assigned" were merged into a unique level of protection categorized as "Unknown".

MPAs resulting from different designation processes can spatially overlap. To avoid double-counting MPA coverage that protect the same area, we only kept the designation providing the highest level of protection for a given area. To do so, we created a unique vector layer for each IUCN level of protection by extracting the corresponding polygons from the original vector layer and merging all polygons. We then subtracted the vector layer of the highest level of protection (Ia) from the vector layer of the second highest level of protection (Ib). We then merged vector layers of Ia and Ib MPAs and subtracted it from the vector layer of the following highest level of protection (II), and so on. Finally, we merged the vector layers obtained, resulting in non-overlapping MPA polygons of the highest levels of protection for a given area.

We used bathymetric data from the GEBCO raster layer[37] to assess the depth distribution of protection coverage. The depth distribution of MPAs and OECMs was obtained by extracting values from the bathymetric raster for each MPA and OECM polygon and summing the area of cells corresponding to the same depth realm.

This layer used a polar projection (EPSG:4326) and thus the area covered by each cell size varied with latitude. The area of each raster cell was calculated using the following formula:

$$\text{cell area} = \text{cell height} * \text{cell width} \qquad (1)$$

The height of a cell is constant and equal to:

$$\text{cell height} = \text{resolution in degrees} * \text{minutes per degree} * \text{meters per minute}$$

with resolution in degrees = 0.004166; minutes per degree = 60' and meters per minute = 1852 m. This gives a cell height of 463 m.

The width of a cell is equal to:

$$\text{cell width} = \frac{\text{earth perimeter (latitude)}}{\text{nb of cells}} \qquad (2)$$

with nb of cells = 86,400.

The perimeter of the Earth at a given latitude was calculated as:

$$\text{earth perimeter (latitude)} = 2\pi R * \cos\left(\text{latitude} * \frac{\pi}{180}\right) \qquad (3)$$

with $R = 6378$ km.

To verify our calculations, we performed a second cell size calculation using the cellSize() function from {terra} and compared the matrix of values obtained from both methods. Results were identical and we chose to keep our formula-based calculation method because of shorter processing time.

The depth realms protected by MPAs or OECMs were determined by the bathymetry of protected cells. The benthic depth realm protected by an MPA or OECM cell corresponded to the bathymetry of that cell, and the pelagic depth realms protected corresponded to all realms occurring between the surface and the seafloor. For example, if a protected cell had a bathymetry of 3000 m, it was assumed to protect the upper bathyal (benthic realm) and the epipelagic, mesopelagic, and bathypelagic (pelagic realms).

We calculated the proportion of benthic protection existing for each benthic realm $i$ as follow:

$$\% \text{ benthic proctected}_i = \frac{\text{benthic area protected}_i}{\text{total benthic area}_i} * 100 \qquad (4)$$

We performed the same calculation for pelagic depth zones.

## Three-dimensional distribution of fishing pressure

For the fishing data, we used the most recent (2.0) version of fleet daily fishing activity[15] from Global Fishing Watch (GFW) at the highest available resolution (0.01 degree). GFW collects data from publicly available automatic identification system (AIS) and vessel monitoring systems operated by governments. While only 2% of all fishing vessels carry AIS (mostly large, commercial vessels), those vessels are responsible for 50% of the fishing in economic exclusive zones (EEZ) and 80% of the fishing in the high seas[15]. This dataset was chosen over the FAO global fishing catch dataset because fishing pressure on marine ecosystems is better captured by fishing effort (hours km$^{-2}$) than catch (ton km$^{-2}$) data. We analyzed fishing data from the three most recent years available on the GFW as of the 20th May 2022: 2018 to 2020. We found a strong correlation between the number of fishing hours by gear type and bathymetry across these three years (tau = 0.96, $p$-value $< 10^{-16}$) demonstrating that results were stable across years. In our figures, we represented values from the 2019 dataset. The inter-annual variability is represented as a 95% confidence interval in Fig. 4. Files tracking the intensity (fishing hours cell$^{-1}$) and location (latitude and longitude) of daily fleet activities in 2019 were combined into a dataset of 205,656,988 fishing events. Data files were converted to vector shapefiles using spatial coordinates of fishing activities (see Fig. S2) and fishing activities were then attributed to ecoregions and bathymetric ranges. Information on vessels' gear type was joined to this dataset using the Maritime Mobile Service Identity (MMSI), a unique identification number for vessels.

We assessed the distribution of fishing pressure across depth realms using two approaches: an approach that estimates the broad impacts of fishing activities across depths (hereafter "depth impacted"), and an approach that estimates the depth directly targeted by fishing activities (hereafter "depth targeted"). We considered as impacted by fishing activities all depth realms occurring in the vertical column of that activity, from the surface to the seabed. This assumption is based on empirical and model-based evidence that disrupting one part of the surface-to-seabed continuum has cascading effects on the rest of the continuum because of vertical connectivity processes. Such cascading vertical impacts have been demonstrated between ecosystems as distant as the epipelagic and abyssal benthos[84,85].

To determine the depth realms impacted by fishing activities, we used the bathymetric value at the location of fishing activities and determined the benthic and pelagic realms present in that surface-to-seabed column. As such, only one benthic realm can be impacted by each fishing activity, but several pelagic realms can be impacted at once. We calculated the fishing pressure (hours km$^{-2}$) impacting each benthic depth realm by summing the total hours of fishing activities impacting that benthic depth realm divided by the spatial extent of that depth realm (Fig. 2). Similarly, we calculated the fishing pressure (hours km$^{-2}$) in each 3D realm (Fig. 4) by summing the total hours of fishing activities impacting a benthic depth realm within a given ecoregion and dividing it by the spatial extent of that 3D realm (km$^2$).

To determine the depth realms targeted by fishing activities, we relied of the depth range of fishing gears associated with each fishing activity. This method was only achievable for gear types that discriminate pelagic and benthic activities (Table S2). We considered that benthic activities targeted the benthic depth realm corresponding to the bathymetry of the fishing location. We considered that pelagic activities only had a direct impact on pelagic realms within the depth range of the fishing gear used, and within the depth range (i.e., the bathymetry) of the location fished. For example, a pelagic fishing gear with a depth range of 30–350 m operating at a location of 400 m depth was considered to target both the epipelagic (0–200 m) and mesopelagic (200–1000 m) realms, but if the same vessel operated at a location of 150 m, it was considered to only target the epipelagic. To determine the depth range of pelagic gears, we reviewed technical descriptions in published and gray literature (e.g., NOAA, MSC). We checked the information collected by consulting fisheries experts. The depth range of each fishing gear obtained from this literature review is summarized in Table S3.

Three gear types used in the GFW did not discriminate benthic and pelagic activities: "purse seines", "trawlers" and "fishing", which represented 62% of the total fishing hours registered. To increase our ability to discriminate benthic and pelagic activities, we contacted managers of the GFW database to access further details on registered trawler vessels, the gear type combining the most unspecified (pelagic vs. benthic) fishing hours. This additional data allowed us to further distinguish between bottom and midwater trawlers and assign an additional 17% of the total fishing hours to benthic or pelagic activities, bringing the total proportion of fishing activities for which the impacted depth could be determined from 38% to 55%. The remaining 45% of fishing hours were categorized as targeting an "unspecified" depth (Fig. 4).

### Definition of conservation priority profiles
Relation between protection coverage and fishing pressure was tested using a linear regression model (Pearson method) from the {ggplot} function suite 'ggpubr'.

Conservation priority profiles were defined based on the fishing pressure and protection coverage occurring within each 3D realm. Four categories of fishing pressure were defined using the quartiles of fishing pressure (hour km$^{-2}$ year$^{-1}$) calculated in section 2.3 for 3D realms. Four categories of protection coverage were defined based on the progress towards the achievement of conservation targets. This was calculated first using the CBD target of 30% of total protection coverage by 2030, and then using the target of 10% of high protection coverage by 2030 (e.g., ref. 67). For the first assessment, we distributed 3D realms among four categories of protection coverage: 0–10%; 10–20%, 20–30% and >30%. For the second assessment, we distributed 3D realms among four categories of IUCN Ia/Ib category coverage: 0–2.5%, 2.5–5%, 5–10%, and >10%. For both assessments, we attributed a score to each 3D realm from 1 to 16 reflecting their position in these 4 × 4 categories (Fig. 5A).

We then simplified these 4 × 4 categories into 2 × 2 categories using as thresholds the median value of fishing pressure and 20% of total protection (resp. 5% of Ia/Ib protection coverage), which correspond to halfway progress from 2020 to 2030 global conservation targets. The four resulting categories defined different conservation priority profiles for 3D realms: "highest priority" for above median fishing and below halfway completion of conservation targets; "lowest priority" for the opposite scenario; "fills conservation gaps" for below median fishing and below halfway completion of conservation targets and "mitigates fishing pressure" for the opposite scenario.

### Software, R packages, and scripts
We carried out all vector and raster operations using the {sf}[86] and {terra}[87] packages under R Core Team (2021)[88]. Figures were created using {ggplot2}[89] and {tidyr}[90], and edited using Illustrator ®. QGIS ® was used for the preliminary visualization of data and production of Supplementary figures.

### Reporting summary
Further information on research design is available in the Nature Portfolio Reporting Summary linked to this article.

## Data availability
All data generated in this study are provided in the Supplementary Information/Source Data file. All datasets used in this study are available in open access at the websites of the institutions detailed in our methods.

## Code availability
The codes used to perform this study are available in the following open-access Zenodo repository: https://zenodo.org/records/10246615[91].

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

## Acknowledgements

We thank Jaeyoon Park, Tyler Clavelle and Kristina Boerder for guidance on our analyses of the Global Fishing Watch data, as well as additional datasets to discriminate benthic and pelagic fishing activities. We thank Petri Suuronen and Ray Hilborn for fruitful discussions on the depth range of fisheries. We thank Irene Salinas Akhmadeeva, Sarah Yerrace, and Lauric Thiault for contributing towards the design of Fig. 1. J.C. would like to thank Fondation de France (MultiNet), Biodiversa (METRODIVER and MOVE), and the European Commission (MARHAB) for financial support. J.J. was awarded a grant by Fulbright France and the University of Washington School of Aquatic and Fishery Sciences.

## Author contributions

Conceptualization: J.J. and J.C. Methodology: J.J., J.C., and C.L. Investigation: J.J. Visualization: J.J., J.C., and C.L. Funding acquisition: L.T. and J.C. Supervision: J.C. and L.T. Writing—original draft: J.J. Writing—review & editing: J.J., J.C., L.T., and C.L.

## Competing interests

The authors declare no competing interests.
