## [Peer Review File · Nature Communications]

3D ocean conservation: Fisheries reach deep but marine protection remains shallowReviewers' Comments:

Reviewer #1:

Remarks to the Author:

The manuscript by Jacquemont and colleagues is a very well written summary of the results of an analysis that has not been done before. In short, the compare the areas covered by marine protected areas with the world's footprint of fisheries. Their most important finding (from my perspective) is that "the protection coverage of 3D realms (log transformed) was negatively correlated with fishing pressure (Figure 5), indicating a large bias of ocean conservation towards least impacted areas" (page 8). This is a conclusion that needs to be highlighted everywhere in the manuscript. I would at least edit the abstract to make it more clear, and I strongly suggest the authors to highlight this whenever they give interviews about this paper when it's published.

The paper does highlight the fact that biologically important areas (mesophotic and rariphotic) are heavily impacted and under protected. This is another important finding that may trigger higher protection of those depths. So the article is timely and its publication is needed.

I don't particularly like the focus on the high seas, as I think the impacts there are still too small and the area is too large, so I would tone that down a bit. But that's just my personal take, so do with it what you will.

A point that I do think should explicitly be made, perhaps in the conclusions section, is that protecting mostly high seas to reach the (completely arbitrary in my view) 30% is a big mistake. And that's where we are heading.

I caught one misspelling, UICN (should be IUCN) in figure 5. Other than that, congratulations on a great manuscript. The analyses all look fine to me, and there is not much else you can do with the data at hand.

Reviewer #2:

Remarks to the Author:

The premise and purpose of the paper are excellent. The main idea is that ocean conservation has traditionally been 2 D but needs to be conceived in 3D to protect the ocean, in particular deep waters. The title with its focus on depth of fisheries is part of the story but maybe not the full one. Please clarify more clearly why consider protecting waters below 1500 m where most fishing ends. Fishing is only one of multiple uses and stressors that affect deep-ocean ecosystems.

The paper has many excellent features that highlight the disconnect between fishing and current ocean protection. The findings will provide important baseline information for spatial planning to achieve 30x30, including in areas beyond national jurisdiction. However, many of the analyses seem complicated and sometimes non-intuitive. In some cases the terminology used does not accurately represent the concept (detailed below). The regional designations of coastal and offshore ecoregions are confusing and somehow perpetuate the false idea that EEZs don't host much of the deep sea. If coastal ecoregions cover only continental shelves (which traditionally extend to 200 m depth) how is any actually bathyal or abyssal? Or do they extend 200 miles from shore?

General Consideration. I find it troublesome to design protection priorities based on current/recent fishing effort without mentioning climate change. You acknowledge that fish may change depth with climate but they also will migrate horizontally. At the very least there should be a mention of how climate change-induced redistribution of species (e.g., will low-m and high- use areas remain so?) will affect your prioritization and whether there is a way to build this in to planning.

Below are considerations and suggestions for clarification.

Summary

Line 20-21 this terminology makes no sense – mesopelagic fishing peaking in areas of abyssal depths. Change to 'in areas overlying abyssal depths'.

Line 30 – can you give some examples of OECMs – are you considering VMEs? APEIs? PSSIs? Or are these on the IUCN categories? Please clarify for those not familiar with the concept.

Line 33 These agreements also bring new opportunities (not just challenges) – and for purposes of this paper it is better to highlight these.

Line 41-49. Two points to consider.

1. It is important to point out that life does not exist very far above the surface on land (maybe 100 m? 1 km?) but life exists for thousands of meters above the seafloor with an average depth of the ocean at 3.8 km. This means that most of the habitable volume on the planet is in the (deep) ocean.
2. UNCLOS has divided international waters into two regimes – the Area (seafloor) managed for minerals and the High Seas (water column) managed for fisheries. BBNJ tackles all the rest.

Line 51 deep sea is only hyphenated when used as a double adjective (e.g. deep-sea fishing) Further, you cannot list deep sea as an ecosystem parallel to mesopelagic reefs or deep coral reefs (which I would argue are the same thing). Do you mean deep-sea sediment? Seamounts? Canyons? Hydrothermal vents? Methane seeps? Abyssal plains? There are many different deep-sea ecosystems present.

Line 67 do you mean avoid or minimize fisheries impacts? Or mitigate in the sense of recovery?

Figure 1. Where is the hadopelagic? This is a valid zone that appears in most textbooks. All trenches have a significant water column.

Line 91-2. Please replace 'most' with a percentage. Are you saying most of the abyssal area is in ABNJ? Note that 52% of all EEZ area has a seafloor between 2000 and 6000 m and much of this is abyssal. So this statement can be misleading.

Line 94 Use of the term twilight zone for lower mesophotic and rariphotic (60-300 m?) is probably incorrect. Standard usage of this term by biological oceanographers refers to the twilight zone as 200-1000 m (see Wikipedia). I suggest not using this term.

Figure 2. This figure has a number of problems.

-I cannot see vertical dashed lines referred to in the legend.

- I do not understand the meaning of distance of depth realm values to benchmarks or how to interpret the lollipops. What forms the benchmark?...

-Most importantly I really think that section D has the legend reversed. As shown you have most of the shallower areas in ABNJ and the deeper areas in the EEZ. Is this correct?

IUCN Categories – Line 88. The first mention of IUCN is on line 88. You need to both write out the full name of the organization – then give the abbreviation. Please explain the categories in the text with reference to the table before you start using them. I had no idea what Ia, Ib, IV etc. were and didn't find this information until I got to the methods.

Fig. 3 - Legend – what is the bottom of the scale assuming the first color is 0-1. 30 to ?? or label > 30. Something is wrong with the white areas in this and other figures. Please clarify how far offshore the coastal ecoregions go – is this 200 miles? Is offshore beyond 200 mi? Or determined another way? I find this terminology very confusing.

Line 126 – replace higher with ‘upper’ bathyal... higher bathyal depths would refer to a larger number but I don't think that is what you mean.

Line 132 Phrasing is confusing – fishing is as high in areas of abyssal depths makes no sense when you say there is no fishing at abyssal depths. All references to this should say ‘in areas overlying abyssal seafloor.’

Line 138 and 144 – ‘Bathymetric distribution’ is not the right term. Refer to depth of seafloor underlying the fishing effort.

Line 142. Do you mean that mesopelagic fisheries have the potential to alter the carbon cycle? I would be very surprised if there is actual demonstration that this has already occurred.

Line 160. Note that deep-sea biologists generally consider the deep sea to start below 2000 m It would be great to know what % of fishing effort occurs at depths > 200 m.

Line 167 – I do not think your study is the first to characterize depths of global fisheries. See Fig. 1 in Morato et al. 2006 (Fish and Fisheries 7: 24-34). This is based on catch but does a good job of illustrating trends in depth. Also Watson and Morato 2013 <https://doi.org/10.1016/j.fishres.2012.12.004>

Line 169 to 170. Change to do not discriminate between..

Line 170-72. Note that the FAOs RFMOs exist separately for demersal fisheries and for pelagic (mostly tuna) fisheries in international waters. Thus management of fisheries and the catch records are distinct for benthic and pelagic in international waters. You did not use their data sets but many papers do.

Line 178 – is drilling about oil and gas? Please be more specific.

Figure. 5

-Change UICN to IUCN in the heading top the left panel.

-Can you explain why protection is on the x axis and fishing is on the y axis? If you considered fishing activity dependent on amount of protection this would make sense... but later in the paper you say this is not the case. I think that discussion needs to be moved up to where you first introduce the negative correlation in Fig. 5. (normally expected) . Since you are trying to argue that high fishing pressure should drive protection I would reverse the axes.

-Please give R2 values with the P. The lines don't look like they explain a great deal of the variance.

Figure 6 assumes static fishing pressure and does not acknowledge other pressures that might occur in bathyal and abyssal waters such as from seabed mining or carbon dioxide disposal technologies.

Line 250 and 253 – the dichotomy between twilight and deep is not appropriate. Typically Twilight is 200-1000 m and deep is below 200 m so one is a subset of the other.

Line 261-2 This is an important point. A perfect example is the Cross Chapter Box on Biodiversity Hotspots in the IPCC Special Report on Ocean and Cryosphere under climate change. None of the hotspots identified occur in the open deep sea.

Line 265. Recognize the deep sea is not fully mapped with respect to bathymetry and mention the Seabed 2030 program goal to fully map the seafloor.

Line 268 – not just depth shifts but also horizontal.

Line 296-8. This statement does not apply to hydrothermal vents and seeps where reliance is not on surface production.

Line 304 - define residual conservation

Line 308 - Area-based conservation implies 2D. Do we need a new term?

Line 320-22. I agree that this is important for ABNJ but remember that 75% of EEZs are in deep water (>200 m) and this is where the largest fishing pressure is.

Line 345 - really should include hadopelagic. If you look up papers in google scholar you will see that this zone (water in trenches) is a real thing with much research.

Line 361 - Introduce Table S1 near the start of the paper.

Line 416 to 418. If you are only tracking 50% of fishing in EEZs what sort of fishing are you missing... is there a particular bias? Can you compare your data set to catch depths?

Figure S1. The colors are generally hard to decipher, especially the very dark color does not distinguish ocean from continents.

Figure S2. Suggest adjusting the slide color scale so more of the fishing activity is visible.

Figure S3. Where is the hadopelagic?

Fig. S4. This is very confusing... How can abyssal sites be such a small percent of most realms. If the first 12 columns are for coastal waters and the last for offshore waters this needs to be labeled and explain the zones. But it seems even then that abyssal systems are underrepresented. Throughout the paper I have found these designations confusing.

Figure S5. What are the left and right panels? Same data with different color scales?

Table S1. Clarify fishing status

Table S2. Dredge fishing entry has a typo

Table S3 why list hadal before abyssal? A. Are you sure there is no abyssal fishing? B. Why no hadopelagic?

Response to reviewers

Reviewer #1

Reviewer #1: The manuscript by Jacquemont and colleagues is a very well written summary of the results of an analysis that has not been done before.

Reply: We deeply thank the reviewer for his/her appreciation of our work, and for pointing out its novelty and clarity.

Reviewer #1: In short, they compare the areas covered by marine protected areas with the world's footprint of fisheries. Their most important finding (from my perspective) is that "the protection coverage of 3D realms (log transformed) was negatively correlated with fishing pressure (Figure 5), indicating a large bias of ocean conservation towards least impacted areas" (page 8). This is a conclusion that needs to be highlighted everywhere in the manuscript. I would at least edit the abstract to make it more clear, and I strongly suggest the authors to highlight this whenever they give interviews about this paper when it's published.

Reply: We thank the reviewer to help us identify key messages. We have now modified the abstract accordingly and will keep that in mind when communicating on our research. The new sentence in the abstract reads as: "Importantly, we found that conservation effort is heavily biased towards areas where least fishing pressures occur, compromising the effectiveness of the marine conservation network globally." (line 22-23)

Reviewer #1: The paper does highlight the fact that biologically important areas (mesophotic and rariphotic) are heavily impacted and under protected. This is another important finding that may trigger higher protection of those depths. So the article is timely and its publication is needed.

Reply: We thank again the reviewer for his appreciation of our study.

Reviewer #1: I don't particularly like the focus on the high seas, as I think the impacts there are still too small and the area is too large, so I would tone that down a bit. But that's just my personal take, so do with it what you will. A point that I do think should explicitly be made, perhaps in the conclusions section, is that protecting mostly high seas to reach the (completely arbitrary in my view) 30% is a big mistake. And that's where we are heading.

Reply: While we believe that emphasizing the importance of our framework and findings in the context of the recent BBNJ agreement is important, we agree that improving the conservation status within national jurisdictions is equally important and that protecting the high seas should not be used to deviate from national targets within National Biodiversity Strategy Action Plans. We have now rephrased the conclusion to emphasize the importance of conservation in EEZs, and tone down our focus on the high seas alone, as suggested: "Such considerations are extremely important for national conservation strategies, as 75% of the world's EEZs consist of deep ecosystems while concentrating the highest levels of fishing pressure. In parallel, the recent conclusion of the High Seas treaty offers a unique opportunity to build upcoming conservation efforts on a revised framework that accounts for the complex and intricately connected three-dimensional ocean space." (line 353-355)

Reviewer #1: I caught one misspelling, UICN (should be IUCN) in figure 5. Other than that, congratulations on a great manuscript. The analyses all look fine to me, and there is not much else you

can do with the data at hand.

Reply: Thank you for catching this mistake. It has now been rectified.

Reviewer #2

Reviewer #2: The premise and purpose of the paper are excellent. The main idea is that ocean conservation has traditionally been 2D but needs to be conceived in 3D to protect the ocean, in particular deep waters.

Reply: We deeply thank the reviewers for his strong appreciation of our work.

Reviewer #2: The title with its focus on depth of fisheries is part of the story but maybe not the full one. Please clarify more clearly why consider protecting waters below 1500 m where most fishing ends. Fishing is only one of multiple uses and stressors that affect deep-ocean ecosystems.

Reply: We agree that other stressors and use affect the deep ocean, but as they are not assessed in this study, we believe that it is more adequate to conserve a title that reflects our study's focus. We chose to focus on fisheries as they are considered the greatest direct anthropogenic threat to marine ecosystems (IPBES, 2019), and because they represent the threat that area-based conservation tools most commonly regulate. The main argument to protect waters below 1,500 m is twofold. First, due to the vertical connectivity across the water column, epipelagic fishing in > 1,500m deep waters will affect indirectly the functioning of these deep water pelagic and benthic systems. Second, fishing activities are targeting deeper and deeper areas, hence protecting 3D areas before fishing occurs is an important strategy for biodiversity conservation. These arguments are explained in the manuscript:

- Line 158-160: *“These indirect impacts occur through vertical connectivity processes, such as migration of organisms, top-down trophic controls or nutrient transfers³⁸⁻⁴⁰; and through by-catch, entanglements, anchoring, fishing debris or ship collisions⁴¹.”*
- Line 327-333: *“This assumes that impacts from human use remain compartmentalized to the depths where they occur, which overlooks numerous connectivity processes across depths. The complex energy, nutrient and population exchanges from the epipelagic to the benthos^{75,76} imply that disturbing one depth realm will likely have cascading effects on any other depth of that system⁴⁰. This is true for shallow but also for most deep ecosystems, which almost exclusively rely on the biomass and productivity originating from epi- and mesopelagic realms^{6,38}.”*

Reviewer #2: The paper has many excellent features that highlight the disconnect between fishing and current ocean protection. The findings will provide important baseline information for spatial planning to achieve 30x30, including in areas beyond national jurisdiction.

Reply: We thank the reviewer for pointing out the implications of our work, which is what guided us.

Reviewer #2: However, many of the analyses seem complicated and sometimes non-intuitive. In some cases the terminology used does not accurately represent the concept (detailed below). The regional designations of coastal and offshore ecoregions are confusing and somehow perpetuate the false idea that EEZs don't host much of the deep sea. If coastal ecoregions cover only continental shelves (which traditionally extend to 200 m depth) how is any actually bathyal or abyssal? Or do they extend 200 miles from shore?

Reply: We agree that this terminology can lead to confusions, and that it was not explained clearly enough in the original manuscript. Coastal ecoregions as defined by Spalding et al., (2007) extend indeed to 200 nautical miles offshore, or to the 200m isobath if the later occurs further. This is why coastal ecoregions include bathyal and abyssal depths, although the area of abyssal depths is very small in coastal ecoregions (see Figure S4). We have now added a clear description of the boundaries of coastal ecoregions:

- in the Method section “*Ecoregions are divided into 11 coastal ecoregions and four “pelagic” ecoregions (Fig. S1). Coastal ecoregions extend to 200 nautical miles (370 km) offshore or to the 200-m isobath where the later occurs further. As such, coastal ecoregions cover all waters shallower than 200 m, but also areas of bathyal and abyssal depths when the later occur within 200 nautical miles from shore.*” (lines 367-371)

- at the first mention of coastal ecoregion in the main text: “*...as well as across most of the world’s coastal ecoregions (extending 200 nautical miles offshore, Figure 3).*” (lines 110-111)

- in Figure 3, 4 and 6., the text “within 200 nautical miles” and “beyond 200 nautical miles” was added to describe coastal vs. offshore ecoregions.

Reviewer #2: General Consideration. I find it troublesome to design protection priorities based on current/recent fishing effort without mentioning climate change. You acknowledge that fish may change depth with climate but they also will migrate horizontally. At the very least there should be a mention of how climate change-induced redistribution of species (e.g., will low-m and high- use areas remain so?) will affect your prioritization and whether there is a way to build this in to planning.

Reply: We thank the reviewer for highlighting this important consideration. Because the spatial units (3D ecoregions) we used for our analyses are extremely large, we believe that most climate-induced shifts in species distribution and in turn in fishing effort distribution would not affect our findings. Even if species migrated by several degrees of latitude, and of tens of meters of depth, they would largely remain in the same 3D ecoregions. However, this would be an important consideration when applying our framework to local contexts, for which a finer spatial resolution would likely be adopted. We have now added the following paragraph in our manuscript to address these points: “*Furthermore, climate-induced shifts in species distributions (Pinsky et al., 2020) and in turn in fishing effort distribution (Palacio Abrantes et al., 2022; Cheung et al., 2010) should be accounted for to ensure that protection is targeted towards areas of high priority now and under future climate conditions. While species and fishing effort redistribution are unlikely to alter the findings of this study given the large spatial extent of 3D ecoregions, this consideration should be reckoned by studies applying this framework at a finer spatial resolution.*” (lines 254-259). The citations we have added in this paragraph are the following:

Cheung, William WL, Vicky WY Lam, Jorge L. Sarmiento, Kelly Kearney, R. E. G. Watson, Dirk Zeller, and Daniel Pauly. "Large-scale redistribution of maximum fisheries catch potential in the global ocean under climate change." *Global Change Biology* 16, no. 1 (2010): 24-35.

Palacios-Abrantes, Juliano, Thomas L. Frölicher, Gabriel Reygondeau, U. Rashid Sumaila, Alessandro Tagliabue, Colette CC Wabnitz, and William WL Cheung. "Timing and magnitude of climate-driven range shifts in transboundary fish stocks challenge their management." *Global change biology* 28, no. 7 (2022): 2312-2326

Pinsky, Malin L., Rebecca L. Selden, and Zoë J. Kitchel. "Climate-driven shifts in marine species ranges: Scaling from organisms to communities." *Annual review of marine science* 12 (2020): 153-179.

Reviewer #2: Below are considerations and suggestions for clarification.

Summary

Line 20-21 this terminology makes no sense – mesopelagic fishing peaking in areas of abyssal depths. Change to ‘in areas overlying abyssal depths’.

Reply: We modified the abstract as suggested.

Reviewer #2: Line 30 – can you give some examples of OECMs – are you considering VMEs? APEIs? PSSIs? Or are these on the IUCN categories? Please clarify for those not familiar with the concept.

Reply: We have not considered these area-based management tools, or any other, as *potential* OECMs. We have only worked on OECMs already declared by countries and reported in the World Database on OECMs, as reviewed in Claudet et al., (2022): “Avoiding the misuse of other effective area-based conservation measures in the wake of the blue economy” *One Earth* 5: 969-974. We have now added a definition of OECM at the beginning of the introduction “*OECMs are geographically defined areas that, unlike protected areas, do not have biodiversity conservation as a primary objective, but still achieve biodiversity benefits from their management, such as bottom-contact fishery closures.*” (lines 35-37)

Reviewer #2: Line 33 These agreements also bring new opportunities (not just challenges) – and for purposes of this paper it is better to highlight these.

Reply: We fully agree that highlighting opportunities is a more powerful approach than only highlighting challenges. We have now added this sentence: “*These agreements bring considerable opportunities for marine conservation, by vastly extending areas that can be conserved and by diversifying the types of governance and sectors that can contribute to conservation.*” (lines 39-42)
We also replaced: “*Addressing these challenges requires (...)*” by “*Realizing the potential of these agreements requires (...)*” (line 45).

Reviewer #2: Line 41-49. Two points to consider.

1. It is important to point out that life does not exist very far above the surface on land (maybe 100 m? 1 km?) but life exists for thousands of meters above the seafloor with an average depth of the ocean at 3.8 km. This means that most of the habitable volume on the planet is in the (deep) ocean.
2. UNCLOS has divided international waters into two regimes – the Area (seafloor) managed for minerals and the High Seas (water column) managed for fisheries. BBNJ tackles all the rest.

Reply: Thank you for these considerations.

1. We have now addressed this point by adding this sentence: “*Unlike on land, life in the ocean spans over a considerable vertical range from the surface to the seafloor, with an average depth of 3,800 m.*” (lines 51-52)
2. We added this sentence to highlight challenges linked to the sectoral driven governance of the ocean: “*Besides, the fragmented nature of ocean governance, with multiple sector- and area-specific regimes, hampers a holistic three-dimensional management of the ocean.*” (lines 61-62)

Reviewer #2: Line 51 deep sea is only hyphenated when used as a double adjective (e.g. deep-sea fishing)

Reply: Thanks. We have corrected the hyphenation of “deep sea” accordingly.

Reviewer #2: Further, you cannot list deep sea as an ecosystem parallel to mesopelagic reefs or deep coral reefs (which I would argue are the same thing). Do you mean deep-sea sediment? Seamounts? Canyons? Hydrothermal vents? Methane seeps? Abyssal plains? There are many different deep-sea ecosystems present.

Reply: We thank the reviewer for pointing this out, we had mistakenly written “mesopelagic reefs” instead of “the mesopelagic”, and we also agree that we cannot list the deep sea as an ecosystem parallel to deep reefs. We have now rephrased accordingly : “*...deep marine ecosystems, such as the mesopelagic, deep coral reefs, and seamounts, are under increasing human pressures...*” (line 64).

Reviewer #2: Line 67 do you mean avoid or minimize fisheries impacts? Or mitigate in the sense of recovery?

Reply: We rephrased to more adequately describe our goal: *“We then test whether marine protected areas (MPAs) and OECMs are appropriately sited to provide protection to areas under highest fishing pressure.”* (lines 80)

Reviewer #2: Figure 1. Where is the hadopelagic? This is a valid zone that appears in most textbooks. All trenches have a significant water column.

Reply: We have now added the hadopelagic pelagic realm in Figure 1.

Reviewer #2: Line 91-2. Please replace ‘most’ with a percentage. Are you saying most of the abyssal area is in ABNJ? Note that 52% of all EEZ area has a seafloor between 2000 and 6000 m and much of this is abyssal. So this statement can be misleading.

Reply: Yes, most of the abyssal area is in the ABNJ. We have replaced “most” with “75%”. (line 107)

Reviewer #2: Line 94 Use of the term twilight zone for lower mesophotic and rariphotic (60-300 m?) is probably incorrect. Standard usage of this term by biological oceanographers refers to the twilight zone as 200-1000 m (see Wikipedia). I suggest not using this term.

Reply: The term “twilight zone” has been used on and off by the mesophotic research community, but we agree that it is not ideal given it is also used by biological oceanographers to refer to the pelagic zone at 200-1000 m. We have removed this term throughout the manuscript and replaced it by “mesophotic and rariphotic”.

Reviewer #2: Figure 2. This figure has a number of problems.

- I cannot see vertical dashed lines referred to in the legend.

- I do not understand the meaning of distance of depth realm values to benchmarks or how to interpret the lollipops. What forms the benchmark?...

-Most importantly I really think that section D has the legend reversed. As shown you have most of the shallower areas in ABNJ and the deeper areas in the EEZ. Is this correct?

Reply:

- There are four vertical dashed lines, one in panel A at $x=0.2$, and three in panel B at $x=0.7$, $x=10$ and $x=30$. We believe that they are dark enough to be easily readable, and wonder if a formatting issue occurred during submission.
- the legend in section D was indeed reversed, we thank the reviewer very much for spotting this mistake. We have now rectified this error.
- we have rephrased the legend of Figure 2 to explain more clearly how to interpret lollipops and benchmarks. Benchmark values correspond to the vertical dashed lines in the figure, standing for 2020 and 2030 CBD targets, global average high protection coverage, and global average fishing pressure. Green and red lollipops indicate positive and negative “status” of depth realms in regard to fishing and conservation averages or targets, i.e., negative if experiencing above average fishing pressure or below average protection coverage. The legend of Figure 2 now reads as follow: *“The four vertical dashed lines represent from left to right: average fishing pressure across depths, average coverage of high protection (MPAs of Ia and Ib IUCN categories) across depths, and the 2020 and 2030 CBD coverage targets. Red and hollow, or green and filled lollipops indicate (i) whether fishing pressure in each depth realm is above or below average fishing pressure across depths, respectively; and (ii) whether the current protection coverage of depth realms is behind or ahead of the average coverage of high protection and of the 2020 CBD target.”* (lines 120-123)

Reviewer #2: IUCN Categories – Line 88. The first mention of IUCN is on line 88. You need to both write out the full name of the organization – then give the abbreviation. Please explain the categories in the text with reference to the table before you start using them. I had no Idea what Ia, Ib, IV etc. were and didn't find this information until I got to the methods.

Reply: We have added the full name of the organization at its first mention, and a reference to Table S1 in that same sentence: “(see Table S1 for details on IUCN categories)” (line 103). Throughout the text, we now specify that categories Ia and Ib “tend to more strictly regulate human use” (line 103).

Reviewer #2: Fig. 3 - Legend – what is the bottom of the scale assuming the first color is 0-1. 30 to ?? or label > 30. Something is wrong with the white areas in this and other figures. Please clarify how far offshore the coastal ecoregions go – is this 200 miles? Is offshore beyond 200 mi? Or determined another way? I find this terminology very confusing.

Reply: We have modified the scale labels to indicate >30 as suggested in Figure 3, and have performed a similar modification in Figure 4. We have now added in Figure 3,4, and 6 text that clarifies the boundaries of coastal vs. offshore ecoregions.

Reviewer #2: Line 126 – replace higher with ‘upper’ bathyal... higher bathyal depths would refer to a larger number but I don't think that is what you mean.

Reply: We thank the reviewer for pointing out this mistake, we replaced ‘higher’ with “upper”.

Reviewer #2: Line 132 Phrasing is confusing – fishing is as high in areas of abyssal depths makes no sense when you say there is no fishing at abyssal depths. All references to this should say ‘in areas overlying abyssal seafloor.’

Reply: We have now applied this suggestion throughout this paragraph.

Reviewer #2: Line 138 and 144 – ‘Bathymetric distribution’ is not the right term. Refer to depth of seafloor underlying the fishing effort.

Reply: We have now removed this formulation throughout this paragraph, including in paragraph titles. Titles now read as “Distribution of fishing activities across space and bathymetry” instead of “Bathymetric distribution of fishing activities”; and “Elucidating the three-dimensional distribution of fishing activities” instead of “From bathymetric to three-dimensional distribution of fishing activities”.

Reviewer #2: Line 142. Do you mean that mesopelagic fisheries have the potential to alter the carbon cycle? I would be very surprised if there is actual demonstration that this has already occurred.

Reply: After re-examining the literature, we toned down our statement to: “Concerns are now being raised that mesopelagic fisheries could even affect carbon sequestration in deep-sea sediments by altering the ocean’s biological carbon pump” (lines 160-162).

Reviewer #2: Line 160. Note that deep-sea biologists generally consider the deep sea to start below 2000 m It would be great to know what % of fishing effort occurs at depths > 200 m.

Reply: We have run a new analysis to determine the % of fishing effort occurring below 200 m (37%) and now include this result (lines 177-178).

Reviewer #2: Line 167 – I do not think your study is the first to characterize depths of global fisheries. See Fig. 1 in Morato et al. 2006 (Fish and Fisheries 7: 24-34). This is based on catch but does a good job of illustrating trends in depth. Also Watson and Morato 2013 <https://doi.org/10.1016/j.fishres.2012.12.004>

Reply: Yes, we acknowledge the work conducted by Morato and colleagues, and have cited both these papers in our manuscript. We agree that the original sentence is incorrect, and have changed it to : “Our study is the first attempt to characterize the 3D distribution of fishing activities” (line 185).

Reviewer #2: Line 169 to 170. Change to do not discriminate between..

Reply: We have modified this sentence accordingly, as well as any other occurrence of this formulation throughout the manuscript.

Reviewer #2: Line 170-72. Note that the FAOs RFMOs exist separately for demersal fisheries and for pelagic (mostly tuna) fisheries in international waters. Thus management of fisheries and the catch records are distinct for benthic and pelagic in international waters. You did not use their data sets but many papers do.

Reply: We thank the reviewer for bringing this point to our attention. Unfortunately, FAOs RFMOs datasets are not spatialized (usually present total catch values across large management units, with no coordinates associated with fishing activities), so while the distinction between pelagic and demersal fisheries is useful, depth cannot be determined. This is why we did not use these datasets (nor the Sea Around Us datasets) for this analysis.

We have expanded our original sentence to reflect these challenges: *“Importantly, about 45% of fishing activities reported in the GFW database do not discriminate pelagic and benthic activities (Fig. 4). Alternatively, datasets that distinguish pelagic and benthic fishing activities, such as those produced by Regional Fisheries Management Organizations, provide catch data across large spatial units, which also prevents from determining the depth distribution of fishing activities. Systematically distinguishing between pelagic and benthic activities and increasing the precision of the spatial information associated with catch data would constitute important steps forward to improve our understanding of the depths targeted by fisheries.”* (lines 187-194)

Reviewer #2: Line 178 – is drilling about oil and gas? Please be more specific.

Reply: Yes, oil, gas, and natural gas liquids. We have now specified “hydrocarbon drilling” (line 253).

Reviewer #2: Figure. 5

- Change UICN to IUCN in the heading top the left panel.

- Can you explain why protection is on the x axis and fishing is on the y axis? If you considered fishing activity dependent on amount of protection this would make sense... but later in the paper you say this is not the case. I think that discussion needs to be moved up to where you first introduce the negative correlation in Fig. 5. (normally expected). Since you are trying to argue that high fishing pressure should drive protection I would reverse the axes.

Reply: We have rectified the misspelling of IUCN, and inverted axes in Figure 5 and in the legend of Figure 6 as suggested.

Reviewer #2: -Please give R2 values with the P. The lines don't look like they explain a great deal of the variance.

Reply: We have now added R2 values along with p-values.

Reviewer #2: Figure 6 assumes static fishing pressure and does not acknowledge other pressures that might occur in bathyal and abyssal waters such as from seabed mining or carbon dioxide disposal technologies.

Reply: We restricted the scope of this study to protection coverage and fishing pressure, as fishing is recognized as the greatest direct anthropogenic threat to marine biodiversity (IPBES, 2019), and because they are one of the threats that marine conservation tools are most readily able to mitigate. We state explicitly in the title of our study and in the title of figures that we only consider fishing pressure. We have now also added two sentences recognizing that other pressures exist and should be accounted for to operationalize our framework: *“Here, we restricted our 3D conservation prioritization to two variables: protection coverage and fishing pressure. While fishing pressure is considered as the main direct anthropogenic threat to marine life, other human pressures could be considered to translate our framework into actionable recommendations, especially given the projected rapid expansion of offshore*

renewable energies, hydrocarbon drilling and deep-sea mining.” (lines 249-254)

Reviewer #2: Line 250 and 253 – the dichotomy between twilight and deep is not appropriate. Typically Twilight is 200-1000 m and deep is below 200 m so one is a subset of the other.

Reply: As stated above, this term has been entirely removed from our manuscript.

Reviewer #2: Line 261-2 This is an important point. A perfect example is the Cross Chapter Box on Biodiversity Hotspots in the IPCC Special Report on Ocean and Cryosphere under climate change. None of the hotspots identified occur in the open deep sea.

Reply: We thank the reviewer for this relevant example to support our point. We have now added this example in our manuscript line 296.

Reviewer #2: Line 265. Recognize the deep sea is not fully mapped with respect to bathymetry and mention the Seabed 2030 program goal to fully map the seafloor.

Reply: We have now added “relying on well-described geophysical features such as depth, *which is underway to be fully mapped by 2030,...*” (lines 297), as well as a reference to the Seabed 2030 program in that sentence (Mayer et al., 2018. "The Nippon Foundation—GEBCO seabed 2030 project: The quest to see the world’s oceans completely mapped by 2030." Geosciences 8).

Reviewer #2: Line 268 – not just depth shifts but also horizontal.

Reply: We only mention vertical shifts in this sentence because we focus specifically on processes for which depth representation can provide a “climate-smart” advantage, and this is not as clearly the case for horizontal shifts. However, we now mention horizontal shifts in species distribution and fishing effort in our discussion of potential impacts of climate change to our prioritization result (lines 254-259).

Reviewer #2: Line 296-8. This statement does not apply to hydrothermal vents and seeps where reliance is not on surface production.

Reply: We have changed the phrasing to “...for *most deep ecosystems, which almost exclusively rely on the biomass and productivity originating from the epi- and mesopelagic realms*” to account for the fact that hydrothermal vents and seeps do not rely on surface production. (line 330)

Reviewer #2: Line 304 - define residual conservation

Reply: We define residual conservation at its first mention: “*it is rather the symptom of residual conservation by which protected areas are placed where they least interfere with human use*” (line 312-313).

Reviewer #2: Line 308 – Area-based conservation implies 2D. Do we need a new term?

Reply: Because we advocate for seafloor to surface conservation strategies, we believe that the term “area-based conservation” is still relevant. We fear that volume-based conservation might incentivize vertically zoned protection. However, complementing spatial indicators with depth indicators could be a relevant shift to improve ecological representation. We have now added these sentences: “*Second, indicators used to track the ecological representativeness of marine conservation networks should include a depth dimension in addition to commonly used 2D units, such as ecoregions¹⁶. The 3D ecoregion typology (depth realm x ecoregion) developed in this study could serve as the framework to set goals and track progress towards 3D ecological representativeness. The number of depth realms considered and their depth limits could be further adjusted at the regional scale to account for local ecological specificities.*” (lines 303-308)

Reviewer #2: Line 320-22. I agree that this is important for ABNJ but remember that 75% of EEZs are in deep water (>200 m) and this is where the largest fishing pressure is.

Reply: We have revised our conclusion to incorporate this important consideration: “*In the wake of a*

global push for more and better ocean conservation to reach Kunming-Montreal GBF targets, the full three-dimensional range of marine biodiversity needs to be represented in proposed conservation networks. Such considerations are extremely relevant for national conservation strategies, as 75% of the world's EEZs consist of deep ecosystems while concentrating the highest levels of fishing pressure. In parallel, the recent conclusion of the High Seas treaty offers a unique opportunity to build upcoming conservation efforts on a revised framework that accounts for the complex and intricately connected three-dimensional ocean space." (lines 353-355).

Reviewer #2: Line 345 – really should include hadopelagic. If you look up papers in google scholar you will see that this zone (water in trenches) is a real thing with much research.

Reply: We have added as suggested “hadopelagic (below 6000 m)” to the pelagic realms and modified the depth range of the abyssopelagic to “3500-6000 m”. (line 383)

Reviewer #2: Line 361 – Introduce Table S1 near the start of the paper.

Reply: Table S1 is now introduced L104.

Reviewer #2: Line 416 to 418. If you are only tracking 50% of fishing in EEZs what sort of fishing are you missing... is there a particular bias? Can you compare your data set to catch depths?

Reply: We recognize biases due to incompleteness of fishing activity datasets, and describe it line 194-198: “*Our results underestimate fishing pressure overlying euphotic to the rariphotic depths because the GFW database only documents vessels with automatic identification systems, which does not capture most tropical small-scale fisheries, especially in the Caribbean, South-West Pacific and Indian Ocean where catches are systematically underreported*”^{22,24}. Unfortunately, catch data suffers from similar limitations. A lot of the world’s small-scale fisheries, especially in tropical areas, are simply unreported.

Reviewer #2: Figure S1. The colors are generally hard to decipher, especially the very dark color does not distinguish ocean from continents.

Reply: We have changed the color scale to allow for an easier identification of ecoregions, and switched continents to a white coloring.

Reviewer #2: Figure S2. Suggest adjusting the slide color scale so more of the fishing activity is visible.

Reply: We have adapted our slide color scale and the spatial grid to make fishing activities more visible.

Reviewer #2: Figure S3. Where is the hadopelagic?

Reply: We have now re-analyzed data taking into account the hadopelagic zone, and present the full result in the new Figure S3.

Reviewer #2: Fig. S4. This is very confusing... How can abyssal sites be such a small percent of most realms. If the first 12 columns are for coastal waters and the last for offshore waters this needs to be labeled and explain the zones. But it seems even then that abyssal systems are underrepresented. Throughout the paper I have found these designations confusing.

Reply: We have now labeled coastal and offshore ecoregions as suggested. Area of depth realms were log-transformed in this figure, which is why abyssal systems might seem underrepresented.

Reviewer #2: Figure S5. What are the left and right panels? Same data with different color scales?

Reply: The left panel is for total coverage by all IUCN categories and OECMs, and the right panel is for high levels of protection only (Ia/Ib IUCN categories). We had forgotten to add this important information in the legend, and have now added it.

Reviewer #2: Table S1. Clarify fishing status

Reply: IUCN categories do not have specific guidelines regarding fishing regulations. This is why IUCN categories cannot be perfectly linked to levels of protection as defined by the MPA guide. As explained in the paper, IUCN categories Ia/Ib tend to correspond to higher levels of protection with stricter fishing regulations, while other categories tend to have relaxed to no fishing regulations.

Reviewer #2: Table S2. Dredge fishing entry has a typo

Reply: Thank you, the typo has been rectified.

Reviewer #2: Table S3 why list hadal before abyssal? A. Are you sure there is no abyssal fishing? B. Why no hadopelagic?

Reply: Abyssal should be before hadal, we corrected this mistake. After a thorough search on current fishing depths and discussion with fishing experts, the lower limit of fishing activities currently occurs at around ~ 3000 m for the deepest bottom trawlers. So to the best of our knowledge, no fishing occurs in the abyssal, and if so it would be extremely rare activities. We now have added “hadopelagic” in Table S3B, although no pelagic fishing currently occurs that deep.

Reviewers' Comments:

Reviewer #1:

Remarks to the Author:

The authors conducted a very thorough review of their manuscript. It reads even better now, and is more focused. They followed all of my suggestions, and I have no further comments.

Reviewer #2:

Remarks to the Author:

I have reviewed the author rebuttal and revised ms and found that the authors have thoroughly and carefully addressed all of the concerns, mistakes, and opportunities pointed out in my original review. I thank them - they now have a very valuable contribution.

There are a very minor few points that remain to consider:

Fig. 2 caption. Lollipop explanation still confusing – perhaps separate the explanation for Panels A and B.

Line 178 is this correct "we found that 37% of total fishing effort overlies depths greater than 300 m"? or do you mean 200 m? as stated in rebuttal letter. Either is ok - but please check what you have is correct and not a typo.

Use of the term High Seas Treaty (in abstract and elsewhere) is misleading, although many use this terminology. 'high seas' is the UNCLOS term for the ocean water column in areas beyond national jurisdiction – but the treaty referred to also includes the seafloor (the Area). I think 'BBNJ Agreement' is more suitable – but the authors prefer High Seas treaty they should at least define this more clearly somewhere. This might seem like a small point but I have already seen state representatives misinterpret the treaty coverage (to not include the seafloor) based on the term 'high seas treaty'.

Abstract - to be more accurate I suggest you say conclusion of the treaty NEGOTIATIONS rather than conclusion of the high seas treaty (it didn't conclude).

Also the authors may be interested in this interactive app which show how much of each EEZ occurs at what water depths.

<https://bit.ly/DeepSeaEEZ> . There are quite a few major fishing countries that have A LOT of abyssal waters in their EEZs (Peru, Japan, Philippines etc.)

REVIEWERS' COMMENTS

Reviewer #1 (Remarks to the Author):

The authors conducted a very thorough review of their manuscript. It reads even better now, and is more focused. They followed all of my suggestions, and I have no further comments.

REPLY: We thank Reviewer 1 for their appreciation of our revision.

Reviewer #2 (Remarks to the Author):

I have reviewed the author rebuttal and revised ms and found that the authors have thoroughly and carefully addressed all of the concerns, mistakes, and opportunities pointed out in my original review. I thank them - they now have a very valuable contribution.

REPLY: We thank Reviewer 2 for their appreciation of our revision.

There are a very minor few points that remain to consider:

Fig. 2 caption. Lollipop explanation still confusing – perhaps separate the explanation for Panels A and B.

REPLY: The explanation of lollipop interpretation has been separated for panels A and B as suggested. The legend now reads: *“Figure 2. Distribution of fishing pressure and conservation efforts across depth realms. (A) Average fishing pressure by fishing gear across depth realms. **Lollipops indicate whether fishing pressure in each depth realm is above (red lollipops) or below (green lollipops) global average fishing pressure.** (B) Protection coverage of MPAs (by IUCN categories) and OECMs across depth realms. **Lollipops indicate whether the current protection coverage of depth realms is behind (red lollipops) or ahead (green lollipops) of the average coverage of high protection and of the 2020 CBD target.**”*

Line 178 is this correct "we found that 37% of total fishing effort overlies depths greater than 300 m"? or do you mean 200 m? as stated in rebuttal letter. Either is ok - but please check what you have is correct and not a typo.

REPLY: We did mean 300 m indeed. We thank the reviewer for pointing this out.

Use of the term High Seas Treaty (in abstract and elsewhere) is misleading, although many use this terminology. 'high seas' is the UNCLOS term for the ocean water column in areas beyond national jurisdiction – but the treaty referred to also includes the seafloor (the Area). I think 'BBNJ Agreement' is more suitable – but the authors prefer High Seas treaty they should at least define this more clearly somewhere. This might seem like a small point but I have already seen state representatives misinterpret the treaty coverage (to not include the seafloor) based on the term 'high seas treaty'.

REPLY: To avoid any confusion, we now added in the introduction: *“a legally binding instrument under the UN Convention on the Law of the Sea to protect and sustainably use marine biological diversity in areas beyond national jurisdiction (BBNJ), **otherwise known as the High Seas Treaty**”* L35-37

Abstract - to be more accurate I suggest you say conclusion of the treaty NEGOTIATIONS rather than conclusion of the high seas treaty (it didn't conclude).

Reply: We added “negotiations” L2 as suggested.

Also the authors may be interested in this interactive app which show how much of each EEZ occurs at what water depths.

<https://bit.ly/DeepSeaEEZ> . There are quite a few major fishing countries that have A LOT of abyssal waters in their EEZs (Peru, Japan, Philippines etc.)

Reply: Thank you for sharing this resource.